# Antibiotic-Resistant Bacteria Isolated from Street Foods: A Systematic Review

**DOI:** 10.3390/antibiotics13060481

**Published:** 2024-05-23

**Authors:** Carmine Fusaro, Valentina Miranda-Madera, Nancy Serrano-Silva, Jaime E. Bernal, Karina Ríos-Montes, Francisco Erik González-Jiménez, Dennys Ojeda-Juárez, Yohanna Sarria-Guzmán

**Affiliations:** 1Facultad de Ingenierías, Universidad de San Buenaventura, Cartagena de Indias, Bolivar 130010, Colombia; carmine.fusaro@usbctg.edu.co; 2Facultad de Ingenierías, Universidad de Cartagena, Cartagena de Indias, Bolivar 130015, Colombia; 3Consejo Nacional de Humanidades Ciencias y Tecnologías (CONAHCYT), Mexico City 03940, Mexico; 4Facultad de Medicina, Universidad del Sinú, Cartagena de Indias, Bolivar 130001, Colombia; 5Facultad de Ciencias de la Salud, Universidad de San Buenaventura, Cartagena de Indias, Bolivar 130010, Colombia; 6Facultad de Ciencias Químicas, Universidad Veracruzana, Orizaba, Veracruz 9430, Mexico

**Keywords:** antibiotic-resistant bacteria, antibiotic resistance genes, street foods, safe food

## Abstract

Street food may be a vehicle of antibiotic-resistant bacteria (ARB) and antibiotic resistance genes (ARGs) to humans. Foods contaminated with ARB entail serious problems or challenges in the fields of medical care, animal husbandry, food industry, and public health worldwide. The objectives of this systematic review were to identify and evaluate scientific reports associated with ARB isolated from various street foods. “Preferred reporting items for systematic reviews and meta-analysis” (PRISMA) guidelines were followed. The bibliographic material covers a period from January 2015 to April 2024. Six electronic scientific databases were searched individually for full-text articles; only those papers that met the inclusion and exclusion criteria were selected. Seventeen papers were included in this systematic review. This study highlighted the wide distribution of ARB resistant to β-lactams and other antibiotics, posing significant health risks to consumers. High resistance levels were observed for antibiotics such as ampicillin, ceftriaxone, and tetracycline, while some antibiotics, such as ceftazidime, clavulanic acid, cefoperazone, cotrimoxazole, doxycycline, doripenem, fosfomycin, vancomycin, and piperacillin-tazobactam, demonstrated 100% susceptibility. The prevalence of ARB in street foods varied between 5.2% and 70.8% among different countries. The multiple resistance of various bacteria, including *Escherichia coli*, *Staphylococcus*, *Salmonella*, and *Klebsiella*, to multiple classes of antibiotics, as well as environmental factors contributing to the spread of antibiotic resistance (AR), emphasize the urgent need for comprehensive approaches and coordinated efforts to confront antimicrobial resistance (AMR) under the “One Health” paradigm.

## 1. Introduction

Street food has emerged as an essential part of the culinary experience and social habits in many regions worldwide [1,2,3]. Street foods and beverages are prepared and sold by vendors in urban sites such as marketplaces, train stations, sidewalk stalls, and public transport stops [4,5,6].

These foods, which often reflect traditions and local cultures, are alternatives to homemade food and in general are more convenient and cheaper than those served at restaurants [7,8]. More than 2.5 billion people worldwide consume street food every day, converting street food micro-industries into a vital sector for the economic development of many towns [9,10,11].

Many street vendors, particularly in developing countries, exhibit bad food-handling practices and low hygienic standards in their daily activities [12,13]; consequently, meals with meat, poultry, seafood, fruits, vegetables, and grains prepared in streets are potential vehicles of microbial foodborne bacteria such as *Klebsiella pneumonia*, *Staphylococcus aureus*, *Salmonella* spp., *Listeria monocytogenes*, and *Escherichia coli* for locals and tourists [14,15]. The pathogens can cause mild or severe disease symptoms that include fever, headache, nausea, vomiting, abdominal pain, and severe diarrhoea [16,17,18].

Antibiotics have contributed immensely to vital therapies, bacterial disease treatments, and foodborne infection cures [19,20]; however, some pathogens have developed AMR strategies to defeat the drugs and survive in different environments [21,22,23]. AR is a natural process that occurs over time when bacteria no longer respond to antimicrobial medicines owing to genetic changes [24,25,26]; consequently, bacterial infections become hard or impossible to treat, increasing the risk of severe symptoms, disability, and death in humans and animals [27].

The global rise in AR, because of the extreme, unreasonable, or severe antibiotics overuse in agriculture, cattle raising, and human medicine [28], poses a significant health issue for the World Health Organization (WHO) and national healthcare services that signal a list of AR “priority pathogens” (ESKAPE list: *Enterococcus faecium*, *Staphylococcus aureus*, *Klebsiella pneumoniae*, *Acinetobacter baumannii*, *Pseudomonas aeruginosa*, and *Enterobacter cloacae*) [29,30].

AR in bacteria is a complex phenomenon influenced by both natural and acquired factors. Bacteria exhibit varying degrees of susceptibility to antimicrobial agents, measured by minimum inhibitory concentration (MIC) [31]. Natural resistance, intrinsic to certain species, arises independently of prior antibiotic exposure and is often attributed to the reduced permeability of outer membranes and efflux pumps [21,31]. Acquired resistance, on the other hand, involves the acquisition of resistance genes via horizontal gene transfer or mutations, leading to altered drug targets, drug inactivation, or active drug efflux [21,31].

One of the primary ways leading to bacterial antibiotic resistance is the limitation of drug uptake. Gram-negative bacteria, with their protective outer membranes rich in lipopolysaccharides (LPSs), naturally impede the entry of certain antimicrobial agents [32,33]. Additionally, bacteria lacking cell walls, such as *Mycoplasma* species, are intrinsically resistant to drugs targeting the cell wall, like β-lactams and glycopeptides [34].

Modification of drug targets is another common resistance mechanism. Bacteria alter components like penicillin-binding proteins (PBPs) to diminish drug-binding efficacy [31]. For instance, *S. aureus* acquires the *mecA* gene, leading to PBP2a production, reducing drug binding. Resistance to ribosomal-targeting drugs can result from mutations, ribosomal subunit methylation, or ribosomal protection, impeding the drug–ribosome interaction [35].

Drug inactivation is achieved through enzymatic degradation or chemical group transfer. β-lactamases, a diverse group of enzymes, hydrolyze β-lactam drugs, making them ineffective [21,31]. Other enzymes transfer chemical groups to drugs, such as acetylation, phosphorylation, or adenylation, diminishing their activity [21,31].

Efflux pumps play a crucial role in resistance by expelling drugs from bacterial cells. These pumps, encoded by chromosomal or acquired genes, actively extrude antibiotics, limiting their intracellular concentration [33]. Bacteria possess various families of efflux pumps, including ATP-binding cassettes (ABCs), multidrug and toxic compound extrusion (MATE), small multidrug resistance (SMR), major facilitator superfamily (MFS), and resistance–nodulation–cell division (RND) families, each with specific substrates and mechanisms [31].

Of particular concern are β-lactamases, which inactivate β-lactam drugs through hydrolysis, leading to widespread resistance, especially among Gram-negative bacteria. Carbapenemases, a subtype of β-lactamases, pose a significant threat, with variants like *Klebsiella pneumoniae* carbapenemases (KPCs) and carbapenem-resistant Enterobacteriaceae (CRE) enzymes exhibiting broad resistance spectra [31].

Street food may be a vehicle of ARB and ARGs to humans [25,36]; foods contaminated with ARB entail serious problems or challenges in the fields of medical care, animal husbandry, food industry, and public health worldwide [37]. Pathogens isolated from street food showed resistance to various antibiotics; for instance, *Escherichia coli* is able to acquire and transfer ARGs [31], while *Staphylococcus aureus* and *Salmonella* spp. isolated from street food and seafoods are resistant to several antibiotics [38].

Safe food, including safe street food, is essential to achieve the Zero Hunger global goal and to improve the life quality and psychophysical health of humans worldwide [39,40]. According to the Food and Agricultural Organization (FAO) “*Food security exists when all people, at all times, have physical and economic access to sufficient safe and nutritious food that meets their dietary needs and food preferences for an active and healthy life*” [41,42].

Despite extensive research in the field of AMR, there remains a remarkable gap in elucidating the intricate interplay between street foods and the evolution of pathogens. A complex approach, both at the national and international levels, based on data derived from recent scientific studies, reviews, and meta-analyses, is required for controlling the spread of foodborne pathogens and promoting food safety. This systematic review addresses the urgent need to close this knowledge gap, as well as to inform strategies regarding the identification of ARB and ARGs. The objectives of this systematic review included the identification and evaluation of scientific reports associated with ARB isolated from various street foods.

## 2. Results

### 2.1. Literature Search

The Preferred Reporting Items for Systematic Reviews and Meta-Analyses (PRISMA) statement flow with the four phases of the literature search (identification, screening, eligibility, and inclusion) is shown in Figure 1. Relevant full-text articles using the following search string: (“Street food”) AND ((Antibiotic) OR (Antibiotic resistance) OR (Antibiotic resistance bacteria) OR (Antibiotic resistant bacteria)), available on six electronic scientific databases were obtained. During the identification phase, 294 publications were recorded. Duplicate articles were automatically removed via a bibliographic management software (Mendeley Desktop Reference Management System 2.111.0), and the remaining 164 papers were screened for title and abstract pertinence. The eligibility of the 133 full-text articles that passed the screening phase was assessed based on present inclusion and exclusion criteria. Finally, only 17 papers were included in this systematic review [43,44,45,46,47,48,49,50,51,52,53,54,55,56,57,58,59].

### 2.2. Basic Characteristics of Selected Studies

The scientific studies were realized principally in developing countries, i.e., Bangladesh [52,53], Burkina Faso [56], Ecuador [59], Ghana [45,49], India [50,51,57], Iran [55], Nepal [44], Nigeria [43,46], and Taiwan [54,58]. Only two studies were performed in Europe, more specifically in Portugal [47] and Poland [48]. Each article provides valuable data about ARB and ARGs isolated from street foods (Table 1).

All selected articles were published in journals belonging to the Scimago Journal Ranking (SJR); more specifically, ten articles were published in Q1 SJR journals [44,45,46,47,48,49,54,55,56,57,58], five articles in Q2 SJR journals [50,51,52,53,55], an article in a Q3 SJR journal [59], and the last one in a Q4 SJR journal [43]. Based on the JBI score rating system tool, 15 of the 17 selected articles were considered as high-quality scientific papers [44,45,46,47,48,49,50,51,52,53,54,55,56,57,58], while the remaining two studies were of moderate quality [43,59]. The information included in the studies covers a versatile collection period. Most studies collected data during the period 2013–2021, accounting for the 82.4% of the literature reviewed, with 14 studies falling in this range [43,44,45,46,47,49,50,53,54,55,56,57,58,59]. Thus, this time range provides a solid view of the evolution of research in this field over the last few years. On the other hand, only three studies did not specify a collection period [48,51,52], representing approximately 17.6% of the publications reviewed.

### 2.3. Data on Antibiotic Resistance in Different Types of Food

Table 2 shows that a large variety of street food including meals (meat, poultry, seafood, fruits, vegetables, and grains) and beverages (water, fruit juices, milkshakes, and coffee) were investigated in the selected studies. Collected samples from various street vendors were weighed, homogenized, and properly cultured in selective media containing antibiotics to encourage the growth of resistant strains. Following incubation, colonies were isolated and subjected to biochemical tests for identification. Additionally, molecular methods were employed to confirm the presence of ARGs. The list of antibiotics used in the selected studies include amikacin (AKN), ampicillin (AM), amoxicillin/clavulanic acid (AMC), amoxicillin/clavulanate potassium (AUG), amoxicillin (AX), azithromycin (AZ), aztreonam (AZT), chloramphenicol (C), clavulanic ceftazidime (CACL), ceftazidime (CAZ), cephalotin (CET), cefixime (CFM), cefoperazone (CF), cefoperazone sulbactam (CFS), ciprofloxacin (CIP), clindamycin (CLI), cotrimoxazole (COT), ceftriaxone (CRO), colistin (CST), cefotaxime (CTX), cefuroxime (CXM), doxycycline (DCN), doripenem (DOR), erythromycin (ERM), ertapenem (ETP), cefepime (FEP), fosfomycin (FOS), cefoxitin (FOX), gentamicin (GEN), imipenem (IMP), kanamycin (K), levofloxacin (LEV), linezolid (Lzd), mecillinam (MEL), meropenem (MEM), nalidixic acid (NA), nitrofurantoin (NFT), norfloxacin (NOR), ofloxacin (OFX), oxacillin (OX), penicillin G (PG), quinupristin/dalfopristin (Q/D), rifampicin (RFP), ampicillin/sulbactam (SAM), streptomycin (SMN), sulfamethoxazole (SMZ), spectinomycin (SPT), sulfonamides (SSS), sulfamethoxazole/trimethoprim (STX), tetracycline (TE), tigecycline (TGC), ticarcillin (TIC), ticarcillin-clavulanate (TIM), trimethoprim (TMP), tobramycin (TOB), vancomycin (VAN), and piperacillin-tazobactam (Zosyn) (Table 2).

The disk diffusion test method has been widely used in the selected studies; only Zurita et al. [59] used the VITEK^®^2 Compact System Microdilution Assay in Broth as an alternative.

Analysis of these studies demonstrates the complexity and diversity of AR in various bacteria that may be associated with food we often consume at many street food stalls. In this regard, *Escherichia coli* has demonstrated a remarkable ability to develop resistance to a wide variety of antibiotics. Several studies have documented this resistance in different strains of *Escherichia coli*. For example, Adeleke and Owoseni [43] identified resistance to β-lactams, while Adhikari et al. [44] observed resistance to specific β-lactams, including *bla_CTX-M_* and *bla_VIM_* [47]. The authors found resistance to multiple classes of antibiotics, including sulfonamides, tetracyclines, phenols, β-lactams, and aminoglycosides [53], and highlighted resistance to a wide range of β-lactams and macrolides, including *bla_TEM_* and *mphA*. In addition, Nikiema et al. [56] reported multifactorial resistance including aminoglycosides, sulfonamides, β-lactams, phenols, tetracyclines, aminoglycosides, and fluoroquinolones. Last, Zurita et al. [59] documented resistance to various β-lactams, along with the presence of genes such as *bla_TEM_*, *bla_CTX-M_*, and *bla_SHV_*. In particular, the study by Adeleke and Owoseni [43] identified the presence of AR *Shigella* strains in several β-lactams, including CXM, AUG, AX, CAZ, and NFT. In the case of *Staphylococcus*, several studies have identified its ability to resist a wide range of antibiotics, including resistance to β-lactams such as CXM, AUG, AX, and CAZ, documented by Adeleke and Owoseni [43], as well as resistance to macrolides, tetracycline, and other β-lactams, according to Chajęcka-Wierzchowska et al. [48], Mesbah et al. [55], Sivakumar et al. [57], and Yang et al. [58]. On the other hand, *Klebsiella* presents resistance to β-lactams, including AM, CTX, FEP, CFS, ETP, ERM, IMP, and MEM, in addition to the presence of resistance genes such as *bla_TEM_* and *bla_CTX_*, as noted in the study by Giri et al. [50].

Gurrurajan et al. [51] also reported resistance in *Klebsiella* to several antibiotics, including β-lactams, with genes such as *bla_CTX-M_* and *bla_SHV_*, and aminoglycosides, such as AM, CAZ, CTX, FEP, Zosyn, GEN, AKN, and TOB. *Citrobacter freundii* and *Klebsiella pneumoniae*, according to Dela et al. [49], are resistant to β-lactams with the presence of *bla_TEM_* and *bla_CTX_* genes. Especially for *Vibrio parahaemolyticus*, Beshiru and Igbinosa [46] reported resistance to a number of antibiotics, including sulfonamides, tetracycline, trimethoprim, β-lactams, and aminoglycosides. *Stenotrophomonas maltophilia*, according to Lin et al. [54], shows resistance to multiple antibiotics, including aminoglycosides and phenols. In addition, *Enterobacteriaceae*, as described in the same study by Lin et al. [54], exhibits resistance to β-lactams, including *bla_CTX-M_* and *bla_SHV_*, as well as aminoglycosides and tetracycline. These findings highlight the growing concern about antibiotic resistance in these bacteria and the need to address it effectively in the clinical and public health settings.

### 2.4. Susceptibility to Antibiotics in the Selected Studies

Some bacteria are observed to show high resistance, with 100% resistance to antibiotics such as AM, AUG, AX, CAZ, CET, CFM, CFS, CST, CXM, LEV, MEL, NOR, OFX, Q/D, SAM, SMN, SMZ, SPT, SSS, STX, TE, TIC, TIM, TMP, and TOB. Antibiotics such as C, CTX, ERM, GEN, MEM, and NFT are found in the 80–99% resistance range, while the 60–79% resistance range includes FEP, FOX, K, and OX. Some antibiotics show 50% resistance, such as AZ, AZT, CLI, ETP, Lzd, and PG. Finally, in the resistance range of 1–49% are AKN, AMC, AZM, CRO, IMP, NA, RFP, and TGC. In contrast, some bacteria show 100% susceptibility to CACL, CF, COT, DCN, DOR, FOS, VAN, and Zosyn. These data highlight the variability of antibiotic susceptibility in the studies reviewed in this paper (Figure 2).

### 2.5. Reported Prevalence of Antibiotic-Resistant Bacteria in Street Foods

The selected studies indicated different sample sizes that ranged between 20 [47] and 430 food samples [57]. The prevalence of ARB in the analysed street food samples based on molecular identification methods ranged between 5.2% (Taiwanese street foods) [58] and 70.8% (Ghana street foods) [45]. The fourth part of the selected studies reports a prevalence of less than 20.0% in specific street foods of Taiwan [58], India [57], Iran [55], and Burkina Faso [56]. Only three studies reported a prevalence of ARB higher than 50.0%, in Nigerian [43], Portuguese [47], and Polish [48] street foods (Figure 3).

## 3. Discussion

The presence of ARB in street foods and beverages is a growing problem that raises serious concerns for public health worldwide. In this study, AR profiles and associated resistance genes were investigated in a variety of bacteria isolated from samples of street foods and beverages in different countries, revealing a worrying trend of widespread resistance to multiple classes of antibiotics, which poses a significant challenge to food safety.

A notable prevalence of β-lactam resistance genes was found in the studied strains, including *bla_CTX-M_*, *bla_TEM_*, and *bla_OXA_*. Additionally, the presence of AR in bacteria such as *Escherichia coli*, *Shigella*, *Staphylococcus*, *Klebsiella*, *Bacillus*, *Proteus*, *Salmonella*, *Lactobacillus*, *Citrobacter*, *Streptococcus*, *Acinetobacter*, *Vibrio*, *Clostridium*, and *Bifidobacterium* poses a direct threat to consumer health, with risks of foodborne infections. Moreover, resistance genes to other types of antibiotics such as tetracycline, aminoglycosides, macrolides, fluoroquinolones, vancomycin, sulfonamides, and trimethoprim were identified in various bacterial strains.

The section screening antibiotic sensitivity brings attention to the diversity in resistance levels, particularly concerning vital antibiotics like penicillin and cephalosporin. The presence of ARB in many types of foods emphasizes the urgency of addressing this public health issue in food production stages and food handling.

The AR has become a growing threat in various contexts, from poultry farming to environmental pollution [60]. In poultry farming, especially in small unregulated businesses and aquaculture, the widespread use of antibiotics has raised concerns given overcrowded conditions and poor sanitation issues. The widespread presence of ARB in animals, meat, and the environment highlights the presence of significant selection pressure exerted by the misuse and overuse of antibiotics in animal production and human medicine. Similarly, contamination of water sources with antibiotic residues and the presence of ARGs in rural rivers point to anthropogenic influence on the resistome. Furthermore, the global transportation of animals and food has contributed to the spread of AR [60].

A study by Han et al. [61] revealed the significant association between ARGs and environmental factors, such as rainfall, which increases the abundance of ARGs and facilitates their dissemination from the air to the soil. Therefore, the concentration of antibiotics and the presence of heavy metals are also linked to resistance, posing environmental challenges and threats to human health [62].

On the other hand, resistance to antibiotics in the soil–plant system has been related to the application of manure, generating selective pressure on soil microorganisms and contributing to the spread of ARGs in contaminated agricultural lands. The transfer of resistance from soil to plant occurs through various pathways, including crop planting and horizontal gene transfer. Factors such as soil properties and environmental conditions also affect the spread of ARGs in the agroecosystem [63].

Similarly, the widespread presence of ARB and ARGs in water in several regions of Africa raises concerns about the spread of AMR on the continent. Nigeria, for example, stands out as a focal point with numerous reports on the presence of ARGs in aquatic environments, exposing the lack of considerable efforts to address AMR in water, which highlights the need for coordinated actions and effective strategies in Africa [64].

Despite the complexity of AR in food and its global impact, it is essential to also explore initiatives and strategies aimed at mitigating this intricate problem.

In this context, research efforts that seek to develop new antibiotics and measures to control the transmission of resistance stand out, as highlighted by Sun et al. [65]. These advances are essential to address the consequences of resistance on human health, food security, and the environment.

Additionally, it is important to consider the unique perspective provided by Yadav et al. [66], who explored the side effects and therapeutic potential of antibiotics beyond their antibacterial action. This wider approach opens possibilities for harnessing the properties of antibiotics in the treatment of inflammatory diseases and cancer, highlighting the importance of continued research in this field.

However, despite promising advances, significant challenges remain, as noted by Nadgir and Biswas [67], who highlight the intricate causes of AR, from resistance genes to inappropriate use in medicine and agriculture. The severity of the consequences underlines the urgent need to adopt comprehensive approaches under the “One Health” paradigm.

The effectiveness of antibiotic detection methods varies widely, as noted by Singh et al. [68], ranging from microbiological approaches to the use of biosensors with fluorescent nanomaterials. Notably, nanomaterials in biosensors offer a promising alternative to conventional techniques, allowing specific and quantitative detection. This capacity is crucial for the identification and control of AR, representing a significant advancement in the surveillance of this phenomenon.

In the context of AR in food, a worrying presence of resistance is observed in a wide variety of common food products, such as meats and cheeses, found daily on the streets and in food stalls. This diversity in resistance rates raises significant concerns for public health and food safety, exposing a considerable challenge in addressing this global problem.

The study by Harada et al. [69] highlights that AR in food is a growing problem worldwide. The 46% increase in antibiotic consumption rates between 2000 and 2018 highlights the urgent need to address the misuse of these drugs, as it is strongly correlated with AMR. Furthermore, the identification of substantial variations in use at the national and subnational levels highlights the complexity inherent in this global problem.

The study by Kayode and Okoh [70] highlights the worrying presence of *Listeria monocytogenes* in ready-to-eat foods and its resistance to several antibiotics. This finding highlights the critical importance of improving hygiene practices in the food chain to counteract food contamination during processing and prevent the development of multidrug resistance in these bacteria.

Regarding food contamination in China, Yang et al. [71] revealed significant levels of *Salmonella* spp. in ready-to-eat foods. The resistance of these strains to antibiotics, especially older ones such as tetracycline, reflects the impact of antibiotic abuse in animal husbandry in that region. This discovery emphasizes the pressing need to improve hygiene practices and antibiotic use in food production to prevent public health risks.

The study by Zhang et al. [72] delves into AR in bacteria, focusing on *Escherichia coli* in ready-to-eat foods in China. The high resistance to multiple antibiotics and the presence of resistance genes indicate a worrying connection between the indiscriminate use of antibiotics in animal husbandry and resistance in food. Furthermore, the identification of plasmids carrying resistance genes suggests the possibility of horizontal transfer of resistance between strains and sources, amplifying the risk.

Together, these studies highlight the complexity of AR and the need for comprehensive, collaborative approaches at a global level to address this multifaceted problem. From regulation to public awareness of responsible antibiotic use, urgent and coordinated measures are required to mitigate the risks associated with AR in different contexts. These strategies may include promoting appropriate antibiotic use, enhancing hygiene and sanitation practices in food handling and preparation, and implementing surveillance programs to monitor the prevalence of resistant bacteria in food products.

The selected studies, which frequently presented significant differences in the methodological schemes, typology of street food, antibiotics, and bacteria, indicated mixed results and conclusions; these peculiarities did not allow definitive comparisons (meta-analysis).

## 4. Materials and Methods

### 4.1. Reporting

The systematic review of the literature was performed according to the standardized method of Preferred Reporting Items for Systematic Reviews and Meta-Analyses (PRISMA) and the statement guidelines and the checklist of Moher et al. [73]. Appendix A presents the PRISMA checklist for this study.

### 4.2. Search Strategy

Searching the literature, published from January 2015 to April 2024, was carried out by an author (Y.S.G.). Six electronic scientific databases, i.e., ISI Web of Science (Clarivate Analytics), EBSCOhost (EBSCO Industries), EMBASE (Elsevier), Science Direct (Elsevier), Scopus (Elsevier), and PubMed (National Library of Medicine of USA—NLM), were searched individually for the relevant full-text articles using the following search string: (“Street food”) AND ((Antibiotic) OR (Antibiotic resistance) OR (Antibiotic resistance bacteria) OR (Antibiotic resistant bacteria)).

### 4.3. Inclusion and Exclusion Criteria

The inclusion criteria, applied to full texts for assessing their eligibility, were the following: (a) original article or short communication about antibiotic-resistant bacteria isolated from street foods; (b) article published from January 2015 to April 2024; (c) article written in English; (d) study limited to street food and antibiotic-resistant bacteria; (e) article published in peer-reviewed journals inserted in the Scimago Journal Ranking (SJR).

The exclusion criteria, applied to full texts for assessing their eligibility, were the following: (a) abstract not associated to the full article; (b) article published in non-peer-reviewed source of SJR; (c) article not written in English; (d) review of literature or meta-analyses; (e) letter to the editor; (f) data note; (g) study with high risk of bias based on the Joanna Briggs Institute (JBI) tool [74].

### 4.4. Selection of Studies

The identified articles were compiled using Mendeley Desktop Reference Management System 2.111.0; the bibliographic manager tool automatically removes the duplicates.

Subsequently, two authors (Y.S.G. and V.M.M.) independently screened titles and abstracts to eliminate the irrelevant papers. A third author (C.F.) made a final decision when two reviewers differed about the relevance of specific study.

Inclusion and exclusion criteria were applied to full texts to assess the eligibility of the selected published material. Two authors (Y.S.G. and V.M.M.) independently analysed the full-text papers, and only those that met all criteria were finally selected. Disagreements between the two researchers were resolved through consultation with a third author (C.F.).

### 4.5. Data Extraction and Analysis

Article-level data were extracted from each selected paper and summarized/tabulated in an abstraction-analysis matrix. The summarized information was organized in columns with the following subjects: (a) Reference; (b) Country; (c) Quartile; (d) Risk of bias; (e) Collection period; (f) Food type; (g) Type antibiotic used; (h) Method of testing; (i) Antibiotic-resistant bacteria; (j) Antibiotic resistance; (k) Antibiotic resistance gene; (l) Antibiotics; (m) Number Used of antibiotic; (n) Number resistant antibiotic; (o) Number susceptible antibiotic; (p) Percentage susceptibility; (q) Sample size; (r) Positive samples; (s) Prevalence.

### 4.6. Quality Assessment

Risk of bias was assessed using the standardized critical appraisal instrument from the Joanna Briggs Institute (JBI) [74]. The JBI checklist, which consists of nine items/questions, aims to examine in detail the methodological soundness of each study under consideration. Based on a score rating system, the studies were divided into three categories: high quality (scores between 7 and 9), moderate quality (scores between 4 and 6), and low quality (scores less than 4) (Appendix A). Two researchers (Y.S.G. and V.M.M.) separately assessed the risk of bias. Disagreements between the two researchers were resolved through consultation with a third author (C.F.).

## 5. Conclusions

This systematic review highlights the global concern about ARB isolated from street foods, underscoring the complex relationship between culinary practices, public health, and microbial resistance. The comprehensive analysis of studies from various countries reveals the complex picture of antibiotic resistance, emphasizing the urgent need for intervention. The high prevalence of ARB in various street foods is alarming, highlighting potential risks to public health and the importance of implementing rigorous food safety measures. Finally, addressing this multifaceted challenge requires collaborative efforts from governments, health authorities, healthcare providers, and consumers, highlighting the importance of effective food hygiene standards and comprehensive education to ensure the safety of street foods globally.

## Figures and Tables

**Figure 1 antibiotics-13-00481-f001:**
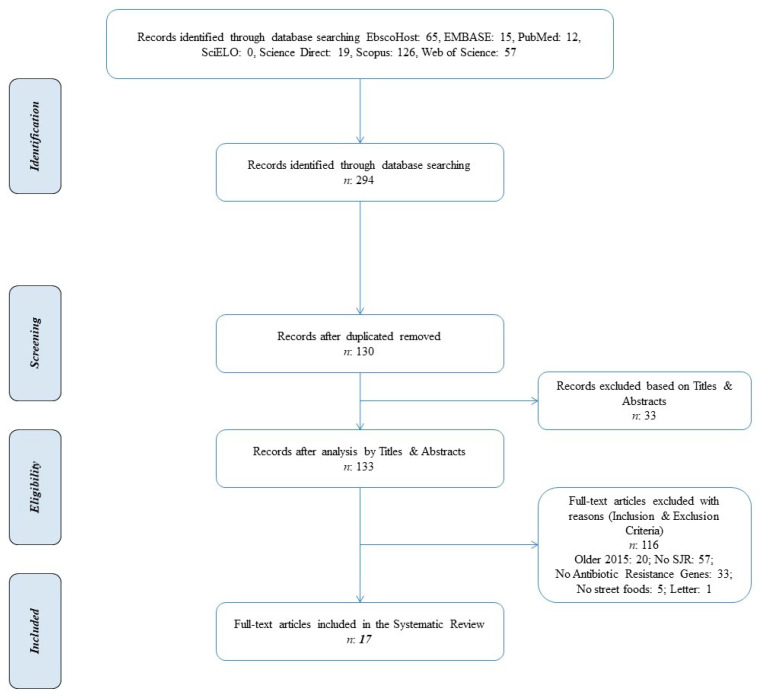
PRISMA flow diagram.

**Figure 2 antibiotics-13-00481-f002:**
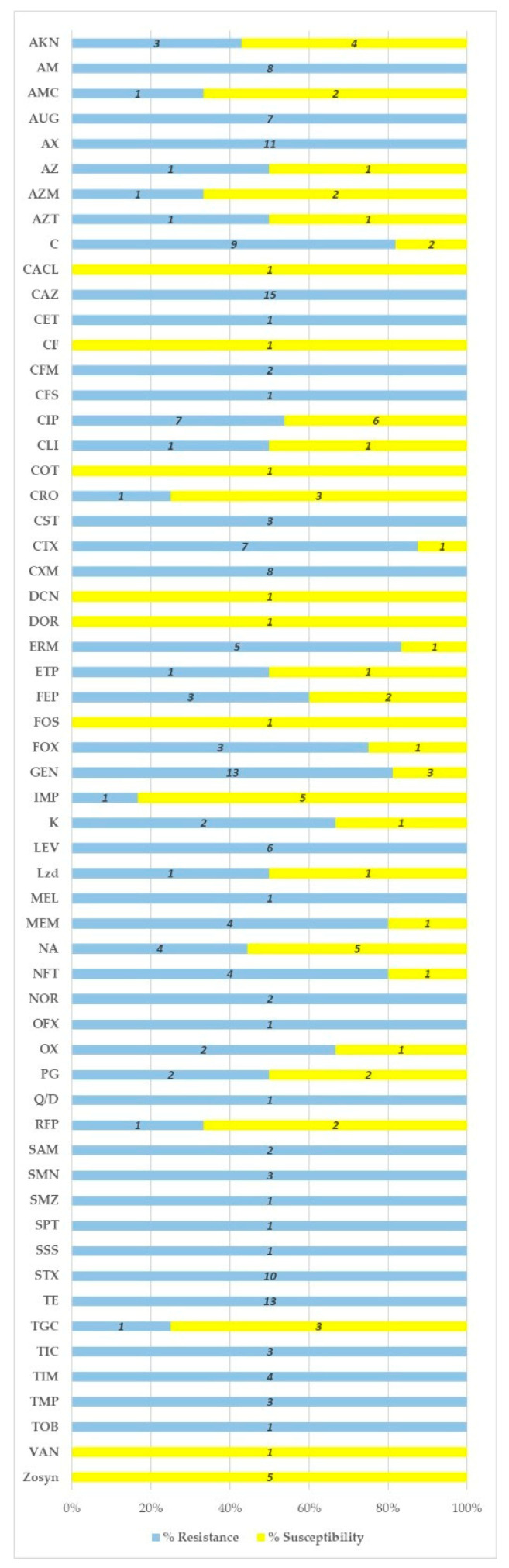
Resistance/susceptibility for each antibiotic used in all included articles. Note—AKN: amikacin; AM: ampicillin; AMC: amoxycillin/clavulanic acid; AUG: amoxycillin/clavulanate potassium (Augmentin); AX: amoxycillin; AZ: azithromycin; AZM: azithromycin; AZT: aztreonam; C: chloramphenicol; CACL: ceftazidime clavulanic acid; CAZ: ceftazidime; CET: cephalothin; CF: cefoperazone; CFM: cefixime; CFS: cefoperazone sulbactam; CIP: ciprofloxacin; CLI: clindamycin; COT: cotrimoxazole; CRO: ceftriaxone; CST: colistin; CTX: cefotaxime; CXM: cefuroxime; DCN: doxycycline; DOR: doripenem; ERV: eravacycline; ERM: erythromycin; ETP: ertapenem; FEP: cefepime; FOS: fosfomycin; FOX: cefoxitin; GEN: gentamicin; IMP: imipenem; K: kanamycin; LEV: levofloxacin; Lzd: linezolid; MEL: mecillinam; MEM: meropenem; NA: nalidixic acid; NFT: nitrofurantoin; NOR: norfloxacin; OFX: ofloxacin; OX: oxacillin; PG: penicillin G; Q/D: quinupristin/dalfopristin; RFP: rifampicin; SAM: ampicillin/sulbactam; SMN: streptomycin; SMZ: sulfamethoxazole; SPT: spectinomycin; SSS: sulfonamides; STX: sulfamethoxazole/trimethoprim; TE: tetracycline; TGC: tigecycline; TIC: ticarcillin; TIM: ticarcillin–clavulanate; TMP: trimethoprim; TOB: tobramycin; VAN: vancomycin; Zosyn: piperacillin–tazobactam.

**Figure 3 antibiotics-13-00481-f003:**
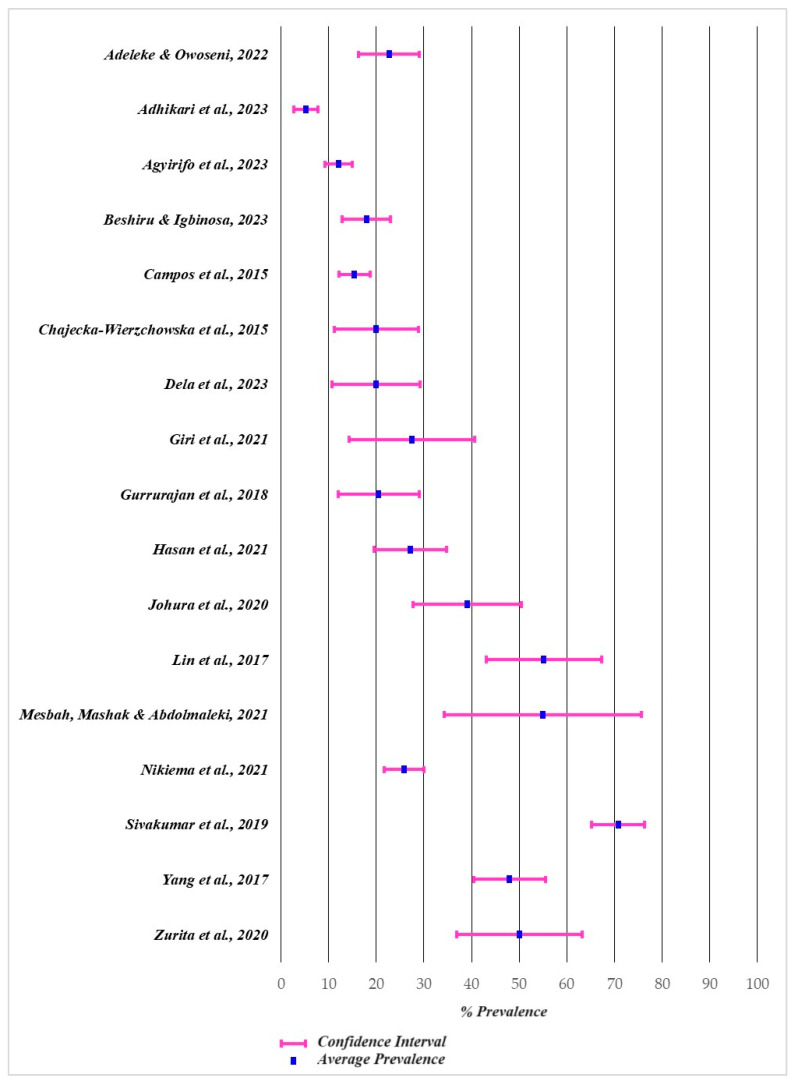
Prevalence of antibiotic-resistant bacteria [43,44,45,46,47,48,49,50,51,52,53,54,55,56,57,58,59].

**Table 1 antibiotics-13-00481-t001:** Basic characteristics of selected studies.

Country	SJR	Risk of Bias	JBI Score	Collection Period	Reference
Nigeria	Q4	Moderate	6/9 66.6%	March–August 2021	Adeleke and Owoseni, 2022 [43]
Nepal	Q1	Low	9/9 100%	September 2019–March 2020	Adhikari et al., 2023 [44]
Ghana	Q1	Low	9/9 100%	January–April 2021	Agyirifo et al., 2023 [45]
Nigeria	Q1	Low	9/9 100%	July 2021–February 2022	Beshiru and Igbinosa, 2023 [46]
Portugal	Q1	Low	7/9 77.7%	January–February 2013	Campos et al., 2015 [47]
Poland	Q1	Low	7/9 77.7%	ND	Chaje˛cka-Wierzchowska et al., 2015 [48]
Ghana	Q1	Low	9/9 100%	January 2019–March 2020	Dela et al., 2023 [49]
India	Q2	Low	9/9 100%	June–October 2019	Giri et al., 2021 [50]
India	Q2	Low	7/9 77.7%	ND	Gurrurajan et al., 2018 [51]
Bangladesh	Q2	Low	7/9 77.7%	ND	Hasan et al., 2021 [52]
Bangladesh	Q2	Low	7/9 77.7%	June 2018	Johura et al., 2020 [53]
Taiwan	Q1	Low	9/9 100%	June–November 2014	Lin et al., 2017 [54]
Iran	Q2	Low	9/9 100%	April–November 2018	Mesbah, Mashak, and Abdolmaleki, 2021 [55]
Burkina Faso	Q1	Low	9/9 100%	June 2017–July 2018	Nikiema et al., 2021 [56]
India	Q1	Low	9/9 100%	September 2015–May 2016	Sivakumar et al., 2019 [57]
Taiwan	Q1	Low	9/9 100%	June–November 2014	Yang et al., 2017 [58]
Ecuador	Q3	Moderate	6/9 66.6%	November 2016–January 2017	Zurita et al., 2020 [59]

Note—JBI: the Joanna Briggs Institute tool; ND: not declared.

**Table 2 antibiotics-13-00481-t002:** Data on antibiotic resistance in different types of food.

Food Type	Type of Antibiotic Used	Method of Testing	Antibiotic-Resistant Bacteria	Antibiotic Resistance	Antibiotic Resistance Genes	Reference
Cooked Street Foods	CAZ, CXM, GEN, CIP, OFX, CFM, AX, NFT, AUG	Disk diffusion	*Escherichia coli*	CXM, AUG, AX, CAZ, NFT	β-lactam (*CTXM*, *OXA*)	Adeleke and Owoseni, 2022 [43]
			*Shigella*	CXM, AUG, AX, CAZ		
			*Staphylococcus*			
			*Klebsiella*			
			*Bacillus*			
			*Proteus*	CXM, AUG, AX, CAZ	β-lactam (*CTXM*)	
			*Salmonella*			
			*Lactobacillus*			
			*Citrobacter*			
			*Streptococcus*	CAZ, CXM, GEN, CIP, OFX, CFM, AX, NFT, AUG	β-lactam (*OXA*)	
			*Acinetobacter*	CXM, GEN, AX, CAZ		
			*Vibrio*	CXM, AUG, AX, CAZ		
			*Clostridium*	CXM, AUG, AX, CAZ		
			*Bifidobacterium*	CXM, AUG, AX, CAZ		
Chutney	STX, CAZ, C, CIP, AZT, AM, GEN, IMP, AX, NA	Disk diffusion	*Escherichia coli*	AM, AX	β-lactam (*bla_CTX-M_*, *bla_VIM_*)	Adhikari et al., 2023 [44]
			*Salmonella*			
Cooked Street Foods (Beans, Fish, Fufu, Kenkey, Pepper sauce, Salad, Soup, Waakye) Vegetables (Cabbage, Carrot, Tomato) Fruit (Apple, Banana, Orange, Pineapple)	AKN, CIP, TE, CRO, CTX, GEN, LEV, NOR, NA, NFT, Zosyn, CF	Disk diffusion	*Staphylococcus*	AKN, CIP, TE, GEN, LEV, NOR	β-lactam (*mecA, bla_TEM_*) Tetracycline (*tetA*, *tetB*) Aminoglycoside (*strA, aacC3*) Macrolide (*ermA*, *ermB*) *Fluoroquinolones* (*acrA*) *Vancomycine* (*vanA*)	Agyirifo et al., 2023 [45]
Agidi jollof, Jollof rice, Fried rice, White ukodo, Soup	AM, TE, C, STX, CIP, CTX, NA, AZ, CAZ, SMN, SAM, GEN, IMP	Disk diffusion	*Vibrio parahaemolyticus*	AM, TE, C, STX, CIP, CTX, NA, AZ, CAZ, SMN, SAM	Sulfonamide (*sul1*, *sul2*) Tetracycline (*tetA*, *tetB*, *tetM*) Trimethoprim (*dfrA1*) β-lactam (*bla_TEM_*) Aminoglycoside (*aadA*)	Beshiru and Igbinosa, 2023 [46]
Hamburgers, Hotdogs	AX, CIP, C, GEN, K, NA, SMN, SMZ, TE, TMP	Disk diffusion	*Escherichia coli*	AX, CIP, C, K, NA, SMN, SMZ, TE, TMP	Sulfonamide (*sul1*, *sul2*) Tetracycline (*tetA*, *tetB*) Phenicols (*floR*, *catA*) β-lactam (*bla_TEM_*) Aminoglycoside (aadA, *strA-strB*) Trimethoprim (*dfrA1*)	Campos et al., 2015 [47]
Cheeses, Cured meats, Smoked fish	ERM, CLI, GEN, FOX, NOR, CIP, TE, TGC, RFP, NFT, Lzd, TMP, STX, C, Q/D	Disk diffusion	*Staphylococcus*	ERM, CLI, GEN, FOX, NOR, CIP, TE, TGC, RFP, NFT, Lzd, TMP, STX, C, Q/D	Tetracycline (*tetL*, *tetK, tetM*) Macrolide (*ermA*, *ermB*, *ermC*) β-lactam (*mecA*)	Chaje˛cka-Wierzchowska et al., 2015 [48]
Ampesi, Banku, Beans, Rice, Salad, Waakye, Soup, Kenkey, Jollof, Spaghetti, Porridge	CRO, STX, Zosyn, TIM, TE, AKN, GEN, CIP, MEM, AZM, C, NFT, NA, CAZ, AMC, ERM, RFP, PG, Lzd	Disk diffusion	*Citrobacter freundii*	STX, TE, NFT, AMC	β-lactam (*bla_TEM_*, *bla_SHV_*)	Dela et al., 2023 [49]
			*Klebsiella pneumoniae*			
Chutney, Dressings, Pickles, Cutlets, Vegetables, Noodles, Pasta, Muffins, Eggs, Chicken, Salami	AM, AKN, AMC, CAZ, CACL, CRO, CTX, CXM, FEP, CFS, COT, C, CIP, ETP, ERM, GEN, IMP, MEM, NA, NFT, Zosyn, TE, TGC	Disk diffusion	*Klebsiella pneumoniae*	AM, CTX, FEP, CFS, ETP, ERM; IMP, MEM	β-lactam (*bla_TEM_*, *bla_CTX_*)	Giri et al., 2021 [50]
Vegetables, Chicken, Samosa, Panipuri water, Bhelpuri	AM, CAZ, CTX, FEP, Zosyn, IMP, GEN, AKN, TOB	Disk diffusion	*Escherichia coli*	AM, CAZ, CTX, FEP, Zosyn, GEN, AKN, TOB	β-lactam (*bla_CTX-M_*, *bla_SHV_*)	Gurrurajan et al., 2018 [51]
			*Klebsiella*			
			*Pseudomonas*			
Phuchka, Eggs	AM, AX, CIP, C, GEN, K, PG, NA, TE, OX	Disk diffusion	*Escherichia coli*	AM, AX, K, TE	Tetracycline (*tetA*)	Hasan et al., 2021 [52]
Juice, Velpuri, Fruit (Guava, Pineapple, Cucumber)	CRO, CET, FEP, CFM, FOS, MEL, TE, STX, LEV, ERM, AZM, IMP, AM, NA, CIP, GEN, C, AZT	Disk diffusion	*Escherichia coli*	CRO, CET, CFM, MEL, TE, STX, LEV, ERM, AZM, AM, NA, GEN, C, AZT	Polymyxins *(mcr*-*1*) β-lactam (*bla_TEM_*) Macrolide (*mphA*)	Johura et al., 2020 [53]
Spring rolls, Noodles, Fruit	CTX, CAZ, C, CST, GEN, LEV, MEM, TIC, TIM, TE, STX	Disk diffusion	*Acinetobacter* spp.	CTX, CAZ, C, CST, GEN, MEM, TIC, TIM, TE, STX, LEV	Aminoglycoside (*aacC1*, *aacC2*, *aacC3*, *aacC4*)	Lin et al., 2017 [54]
			*Pseudomonas* spp.	CTX, CAZ, C, CST, GEN, MEM, TIC, TIM, TE, STX	Aminoglycoside (*aacC2*) Phenicols (*cmlA*)	
			*Stenotrophomonas maltophilia*	CAZ, C, LEV, TIM, STX	Phenicols (*catIII*)	
			*Enterobacteriaceae*	CTX, CAZ, C, CST, GEN, MEM, TIC, TIM, TE, STX, LEV	Aminoglycoside (*aacC4* ) Tetracycline (*tetA*, *tetC*, *tetD*)	
Hamburgers, Chicken nuggets, Salad, Salami, Falafel, Grilled mushrooms, Mexican corn	AKN, GEN, LEV, CIP, CLI, ERM, AZ, PG, DCN, TE, C, STX, RFP	Disk diffusion	*Staphylococcus aureus*	GEN, CIP, ERM, PG, TE, STX	Tetracycline (*tetK*) β-lactam (*blaZ*) Aminoglycoside (*aacA-D*) Macrolide (*ermA*) Fluroquinolones (*gyrA*)	Mesbah, Mashak, and Abdolmaleki, 2021 [55]
Sandwiches	AM, AMC, FOX, CTX, CAZ, FEP, SMN, SPT, GEN, AKN, TGC, K, SSS, TMP, STX, C, TE, NA, CIP, MEM, AZM	Disk diffusion	*Salmonella*	AM, SMN, SPT, GEN, SSS, TMP, STX, C, TE, NA, CIP	Aminoglycoside (*strA*, *strB*) Sulfonamide (*sul1*, *sul2*) β-lactam (*bla_TEM-1B_*) Phenicol (*catA1*) Tetracycline (*tetA*) Aminoglycoside (*aad7*) Fluoroquinolones (*gyrA*, *parC*) Phosphonics (*fosA7*)	Nikiema et al., 2021 [56]
Chicken, Eggs, Milk, Paneer, Fish, Lassi, Salad, Chutney, Masala	OX, FOX, PG	Disk diffusion	*Staphylococcus aureus*	OX, FOX, PG	β-lactam (*mecA*, *blaZ*)	Sivakumar et al., 2019 [57]
Spring rolls, Noodles	ERM, GEN, LEV, OX, TE, VAN	*ND*	*Staphylococcus*	ERM, GEN, LEV, OX, TE	β-lactam (*mecA*) Macrolide (*ermA*, *ermC*) Tetracycline (*tetM*, *tetK*, *tetO*) Aminoglycoside (*aac(6′)Ie-aph(2″)Ia*)	Yang et al., 2017 [58]
Chili pepper sauce, Ceviche, Salad, Cheeses	STX, AM, SAM, FOX, CAZ, CTX, FEP, AKN, GEN, Zosyn, DOR, ETP, IMP, MEM, CIP, TGC, CST	VITEK^®^2 Compact System Broth Microdilution Assay	*Escherichia coli*	STX, AM, SAM, FOX, CAZ, CTX, FEP, AKN, GEN	β-lactam (*bla*_TEM_, *bla_CTX-M_*, *bla_SHV_*)	Zurita et al., 2020 [59]

Note—ND: not declared; AKN: amikacin; AM: ampicillin; AMC: amoxycillin/clavulanic acid; AUG: amoxycillin/clavulanate potassium (Augmentin); AX: amoxycillin; AZ: azithromycin; AZM: azithromycin; AZT: aztreonam; C: chloramphenicol; CACL: ceftazidime clavulanic acid; CAZ: ceftazidime; CET: cephalothin; CFM: cefixime; CF: cefoperazone; CFS: cefoperazone sulbactam; CIP: ciprofloxacin; CLI: clindamycin; COT: cotrimoxazole; CRO: ceftriaxone; CST: colistin; CTX: cefotaxime; CXM: cefuroxime; DCN: doxycycline; DOR: doripenem; ERM: erythromycin; ETP: ertapenem; FEP: cefepime; FOS: fosfomycin; FOX: cefoxitin; GEN: gentamicin; IMP: imipenem; K: kanamycin; LEV: levofloxacin; Lzd: linezolid; MEL: mecillinam; MEM: meropenem; NA: nalidixic acid; NFT: nitrofurantoin; NOR: norfloxacin; OFX: ofloxacin; OX: oxacillin; PG: penicillin G; Q/D: quinupristin/dalfopristin; RFP: rifampicin; SAM: ampicillin/sulbactam; SMN: streptomycin; SMZ: sulfamethoxazole; SPT: spectinomycin; SSS: sulfonamides; STX: sulfamethoxazole/trimethoprim; TE: tetracycline; TGC: tigecycline; TIC: ticarcillin; TIM: ticarcillin—clavulanate; TMP: trimethoprim; TOB: tobramycin; VAN: vancomycin; Zosyn: piperacillin–tazobactam.

## Data Availability

Data are contained within the article.

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
