# Peer review of "Antibiotic-Resistant Bacteria Isolated from Street Foods: A Systematic Review"

_antibiotics, 2024, doi:10.3390/antibiotics13060481_

Round 1

Reviewer 1 Report

Comments and Suggestions for Authors

The manuscript "Antibiotic resistance bacteria isolated from street foods: a systematic review" aims to identify and evaluate scientific reports associated with antibiotic resistance bacteria isolated from various street foods. The study has great importance in the field of AMR research associated with food safety and public health. However, the manuscript requires a few corrections.

Line No.: 42-44: Please check the sentence structure.

The background of the study is not well written. The background of the study is long and has mostly described the antibiotic resistance mechanism. Please mainly focus on antibiotic resistance associated with food-borne pathogens, particularly ready-to-eat foods. What is the knowledge gap of the study? The justification of the study is not well enough. Please mention why the study is needed.

Are there any exclusion criteria related to resistance detection methodology? Did the all included studies follow standard resistance detection methodology?

The quality of Figure 1 is extremely poor. Please make sure all the figures are fully visible and understandable.

Most of the studies are from developing countries, why? Is it an inclusion criterion?

Comments on the Quality of English Language

Minor editing of English language required.

Author Response

Thank you very much for your suggestions.

The rebuttal letter contains all  the corrections.

Reviewer 2 Report

Comments and Suggestions for Authors

Some few suggestions to the authors.

The abstract and introduction was good and exhaustive. I am satisfied with the quality of the presentation.

Result sections

Literature search

The authors should provide an exhaustive breakdown of each of the articles identified from each database search and the sum total from all databases. Also, the reasons for the exclusion of each article after full-text evaluation must be provided and the number of articles based on the reasons provided.

On table 1, I don’t think it is ideal for the authors to express the “risk of bias” on the basis of moderate or low. The JBI quality assessment checklist has a 9 items and it is expressed in 100%. This should be so and not expressing the quality assessment using “moderate” or “low” is not the right thing to do.

The figures presented in this manuscipts were of poor quality from figure 1 to 3.

Discussion

This section is ok

Comments on the Quality of English Language

None

Author Response

(The authors gave the same response as above.)

Reviewer 3 Report

Comments and Suggestions for Authors

The manuscript “Antibiotic resistance bacteria isolated from street foods: a systematic review” is well-presented and highlights the growing concern regarding antibiotic resistance.

I have only minor comments that are mentioned in the PDF.

Comments on the Quality of English Language

English language needs minor editing.

Author Response

(The authors gave the same response as above.)
